# High Precision Control System for Micro-LED Displays

Yufeng Chen [1,2,3], Xifeng Zheng [1,3,*], Hui Cao [3], Yang Wang [1,3], Hongbin Cheng [1,3], Junchang Chen [1,2], Shuo Huang [1,2], Jingxu Li [1,2], Deju Huang [1,2] and Yu Chen [1,3]

1   Changchun Institute of Optics, Fine Mechanics and Physics, Chinese Academy of Sciences,
    Changchun 130033, China; chenyufeng21@mails.ucas.ac.cn (Y.C.); wangy@ccxida.com (Y.W.);
    chenghb@ccxida.com (H.C.); chenjunchang20@mails.ucas.ac.cn (J.C.); huangs@ccxida.com (S.H.);
    lijingxu21@mails.ucas.ac.cn (J.L.); huangdeju21@mails.ucas.ac.cn (D.H.); cheny@ccxida.com (Y.C.)
2   University of Chinese Academy of Sciences, Beijing 100049, China
3   Changchun Cedar Electronics Technology Co., Ltd., Changchun 130103, China; caoh@ccxida.com
*   Correspondence: zhengxf@ccxida.com

**Abstract:** This paper proposes a Field Programmable Gate Array (FPGA)-based control system to implement micro-light-emitting diode (micro-LED) real-time display. The control system includes the interface control, video processing, memory management, image data transmission, control signal generation and correction. Then, we implement the micro-LED real-time display via memory management. We propose the brightness correction to achieve high grey-scale and high uniformity display. The LEDs are mounted on the glass substrate prepared using low-temperature polysilicon (LTPS) technology, and then we find the 24 × 46 pixels micro-LED panel. And the control system has been successfully applied to the panel of glass-based micro-LED displays. A new grey control method is proposed in this work, which can effectively improve the refresh rate of the micro-LED displays. The high grey-scale refresh rate is 2100 Hz, and the low grey-scale refresh rate is 300 Hz. The uniformity of the panel is increased to 85% after brightness correction.

**Keywords:** Micro-LED; FPGA control; memory management; brightness correction; refresh

## 1. Introduction

Display technology permeates daily life, work, and study, and it is essential to modern society. The prevailing technologies in the display industry at the current stage are liquid crystal display (LCD), light-emitting diode (LED) display, organic light-emitting diode (OLED) display, laser display and electronic paper display. LED display includes ordinary LED display, mini-LED display and micro-LED displays. Micro-LED chip size is below 50 microns and densely integrated onto a glass substrate. And micro-LED displays has the advantage of higher brightness, higher luminous efficacy, a faster response time and better color reproducibility than other display technologies [1–4]. These advantages enable it to have broader application value in display scenarios, such as television (TV), mobile phones, wearable devices and augmented reality (AR) [5–9]. Due to its ability to achieve seamless stitching, it will be extensively used in large-size display.

A great deal of research has been carried out in recent years on micro-LED displays, including driving control, uniformity regulation and refresh rate improvement [10–15]. Zou et al. presented a digital method to realize 16 × GGG × 29 micro-LED displays with a 60 Hz refresh rate [13]. Based on a low-temperature polysilicon (LTPS) thin-film transistor (TFT), Samsung used mixed modulation for the first time to directly convert voltage into luminescence time and control different grey-scales [16]. However, the development of micro-LED displays faces some bottlenecks. These include the lack of a dedicated driver IC for micro-LED displays, uneven micro-LED displays due to non-uniform TFT and LED differences and an obvious flicker phenomenon in low grey-scale due to the traditional scanning mode. The traditional scanning mode can achieve high refresh rates in high grey-scale but low refresh rates in low grey-scale. Micro-LED driving can be divided into active

driving and passive driving. The passive driving method is simple and easy to implement, but it is not capable of supporting high-resolution displays. In contrast, active driving technology based on a thin-film transistor (TFT) is used to drive Micro-LEDs. However, micro-LED is a current-driven light-emitting device. The direct current modulation method can lead to color casting problems because of the threshold voltage drift and carrier mobility difference during the TFT preparation. And the brightness fluctuation of micro-LED is caused by differences in the brightness between the micro-LED and the IR drop of TFT. A refresh rate that is too low will produce an obvious flicking sensation, and there will be scanning patterns when the image is captured by photographic equipment.

In this study, we propose a FPGA-based control system that produces pulse width modulation (PWM) data voltages and control signals to precisely regulate the brightness of 24 × 46 glass-based micro-LED displays, achieving 10-bit grey-scale. The high grey-scale refresh rate is 2100 Hz, and the low grey-scale refresh rate is 300 Hz. Additionally, the panel's uniformity can be enhanced through brightness correction. The paper proposes a digital–analogue hybrid method that combines the traditional direct current modulation method with PWM driving to achieve constant current drive control.

## 2. Control System Architecture

Figure 1 shows the micro-LED display's control structure. The control system consists of the interface control, video processing, image segmentation, image data transmission, control signals generation, correction and memory management. The video decoding module uses the high-definition multimedia interface (HDMI) receiver to produce the RGB data and video timing to be sent to image segmentation. Subsequently, the data are sent to the correction model, which includes gamma correction and brightness correction. The corrected data are input to RAM for memory management. One of these four FPGAs serves as the master, and the remaining three operate as dependents. The master FPGA is responsible for global control and part of the PWM data, while the three dependents generate the PWM data.

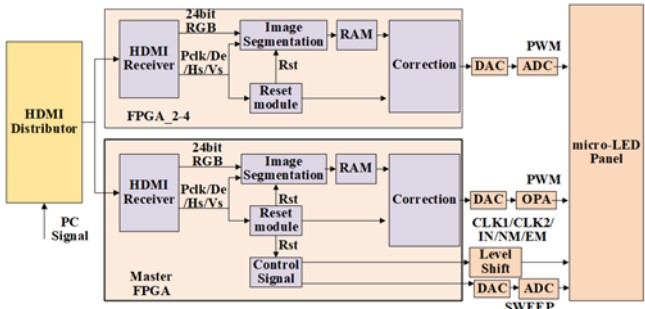

**Figure 1.** The micro-LED displays control structure.

In order to implement the control of the 24 × 46 array, the number of I/O pins of the FPGA needs to be calculated. This process involves adding the number of pins necessary for 10-bit analog voltage signal generation to those required for control signal generation. The number of I/O pins required to generate the 10-bit analog voltage signal can be estimated as follows: m1 = 3 × 10 × n. Here, n is the number of columns in the micro-LED display's array. We can determine the number of I/O pins by substituting n = 24 into the above equation: m1 = 3 × 10 × 24 = 720. The control signals consist of CLK1, CLK2, IN, NM and EM. It should be noted that the chip used to generate these signals requires 2 I/O pins. We estimate that the number of I/O pins required to generate the control signals is 10: m2 = 3 × 5 = 10. And then the total number of I/O pins can be calculated as follows: m0 = m1 + m2 = 720 + 10 = 730. But as the conventional FPGA do not have a sufficient number of I/O pins, we presented a new architecture to implement the 24 × 46 Micro-LED displays. Therefore, we proposed a new architecture to implement 24 × 46 Micro-LED displays using an XC7A200T FPGA. The panel is implemented by stitching the video data.

A state machine is designed via Verilog programming (Vivado 2019.1) to realize video interception. Each FPGA intercepts a portion of the micro-LED displays area. The CLK1, CLK2, IN, NM, EM, SWEEP and 1–6 columns of PWM data are generated by the master FPGA. Similarly, the 7–12, 13–18 and 19–24 columns of PWM data are generated by the FPGA_2, FPGA_3 and FPGA_4. The 4 FPGA are spliced together in the manner shown in Figure 2.

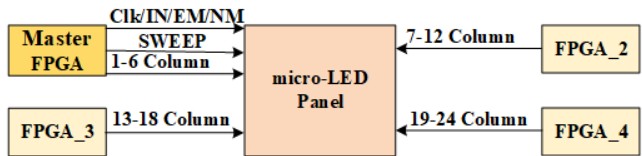

**Figure 2.** The control scheme of four FPGAs.

Figure 3 shows the timing diagram of the micro-LED, and Table 1 shows the voltage range of these signals [12,13,16,17]. These control signals are generated by the control system. The control signals, namely CLK1, CLK2, IN, NM, EM and comparison SWEEP, are generated by the generation module.

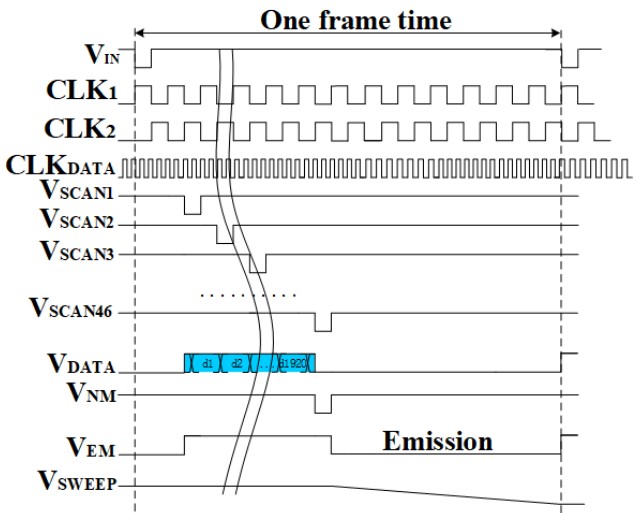

**Figure 3.** The timing diagram.

**Table 1.** Design parameters of the control signal.

| Signal | Value (V) |
| --- | --- |
| CLK1 | −8~8 |
| CLK2 | −8~8 |
| IN | −8~8 |
| NM | −8~8 |
| EM | −8~8 |
| SWEEP | 0~9 |
| DATA | 0~8 |

The CLK1, CLK2 and IN are outputted to the GOA circuit to generate the scan signals row by row. The NM and EM are the timing control signals used for the TFT. Square wave signals of the desired voltage range are output by controlling the single-pole double-throw (SPDT) switch chip. Figure 4 demonstrates the control of the desired voltage range of the control signals through the SPDT switch.

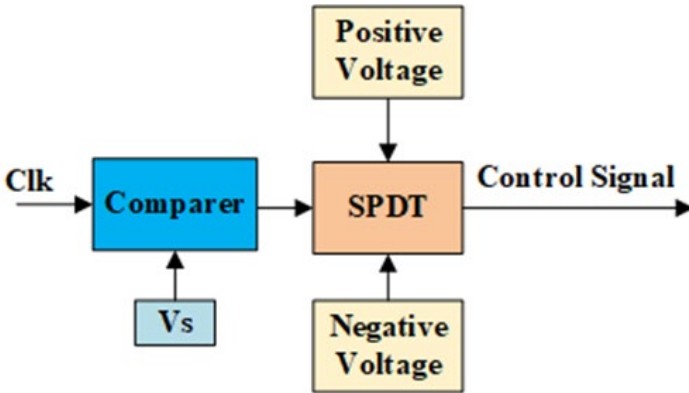

**Figure 4.** The producing method of CLK1/CLK2/IN/NM/EM.

Figure 5 shows the pixel circuit with the analog PWM operation [16,17]. When $PWM + \Delta SWEEP > VREF$, the micro-LED starts to emit light. Afterwards, $SWEEP$ will gradually decrease, and if $PWM + \Delta SWEEP < VREF$, the micro-LED ceases emitting light. We can assume that $PWM + \Delta SWEEP = VREF$; thus, the $SWEEP$ signal change relationship equation can be expressed as follows:

$$\Delta SWEEP = VREF - PWM \tag{1}$$

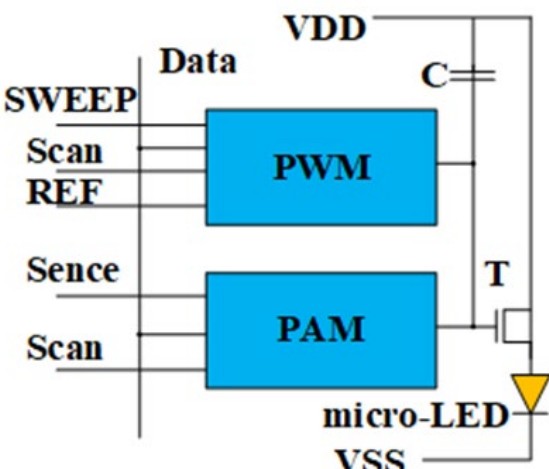

**Figure 5.** The proposed PWM pixel circuit.

Different video data have different PWM signals that require different lighting times; the lighting time can be summarized as follows:

$$T = \frac{|V_{REF} - V_{PWMD}|}{V_{SWEEP}} \times T_{SWEEP} = \frac{\Delta V_{SWEEP}}{V_{SWEEP}} \times T_{SWEEP} \tag{2}$$

Therefore, the grey level of the display is affected by the slope of the SWEEP amplitude drop. The counter and comparator are used to generate a 12-bit decreasing series, and then a 12-bit DAC is used to output a 12-bit precision ramp signal. Then, the ramp signal is stepped up to the drive voltage range through an amplifier circuit, thus achieving a high-precision SWEEP signal output, as shown in Figure 6.

The 8-bit RGB video data read from RAM are fed into correction model and converted into 10-bit data via gamma transformation. Then, the 10-bit data are fed into 10-bit DAC and ADC to obtain the grey-scale voltage, also known as PWM data, as shown in Figure 7.

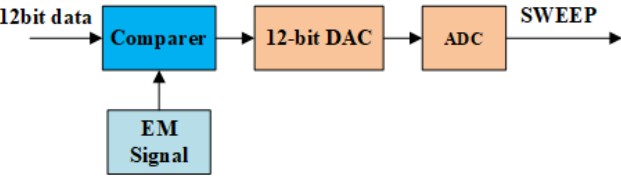

**Figure 6.** The producing method of SWEEP.

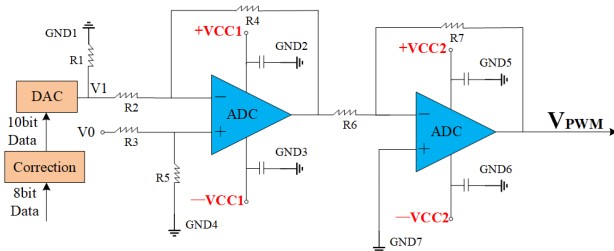

**Figure 7.** The producing method of PWM.

The PWM data voltage can be expressed as follows:

$$V_{PWM_i} \approx \left( 32 \cdot V \cdot R_1 \cdot i \Big/ 1024 \cdot R_2 \right) \cdot (V_1 - V_0) \cdot \frac{R_4}{R_5} \cdot \frac{R_7}{R_6} \tag{3}$$

where the V is the voltage at terminal voltage, $R_1$ the is equivalent input load resistance and $R_2$ is the external resistor of the DAC. Through digital-to-analogue conversion and two-stage amplification, we obtain the PWM data required to achieve real-time display.

Differences in brightness between the micro-LED and the IR drop of TFT may cause inconsistent brightness of the micro-LED. To improve the quality of micro-LED displays, the panel must undergo correction, including gamma correction and brightness correction.

Since luminance correction leads to a loss of grey-scale on the display, we implement front-end video correction. As Figure 8 shows, the RGB data are read out from the RAM for inverse gamma transformation, and the processed image data are corrected by the multiplication module. The processed image data undergo correction through a multiplication module, using CoefR, CoefG and CoefB as correction parameters for the RGB corresponding to their respective image data. For every LED display pixel, there are three correction parameters. And the 10-bit image data of each primary color are multiplied by the corrected parameter values after quantization. The resulting calculations are output to the DAC, and then we can identify the data voltage.

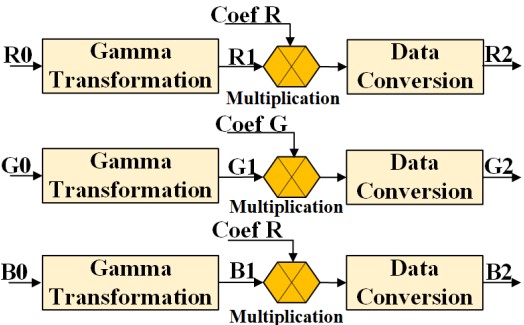

**Figure 8.** Brightness correction.

To guarantee synchronization between the data read from the dual-port RAM and the display timing of the panel, we executed memory management, as shown in Figure 9. The dual-port RAM is used to accomplish both the writing and reading of video data. The

writing address was regulated to choose the particular section of dual-port RAM for writing data, while the reading address was controlled to select which section of the dual-port RAM used to read data. When RAM1 is written, the data in RAM2 are read; when RAM2 is written, the data in RAM1 is read. The data written are serial, while the data read are parallel, meaning that the data readout frequency is decreased through down-sampling to match the display timing of the micro-LED. The pixel number of the micro-LED is 24 × 46, and the input data are serial, meaning that the width of the input port is set at 30. Considering that both RAMs continuously read and write, the depth of the input port is set at 294. The output data are parallel, and only one row of data is read at a time, meaning that the width of the output port is 960. The data are written with a 2 K pixel clock, which is much faster than the pixel clock of a micro-LED panel (24 × 46), meaning that the frequency of the reading data is reduced via down-sampling to match the display timing of the micro-LED. The 10-bit data read from the RAM are fed into 10-bit DAC and ADC to obtain grey-scale voltage, also known as PWM. Four FPGA are used for RGB data image stitching. The global reset is implemented via the vertical synchronizing signal (VSYNC) in order to resolve the desynchronization of data. And the VSYNC is generated via the control system at regular intervals, ensuring no accumulation of errors and guaranteeing the synchronization of the display area controlled by each FPGA.

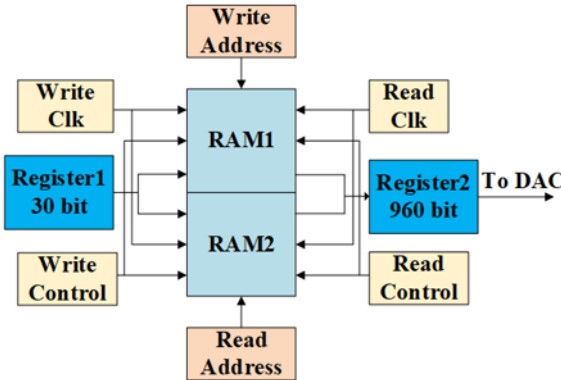

**Figure 9.** The data control with the dual-port RAM.

A new grey-scale control model is presented in Figure 10. The display flicker is reduced by applying the time-slice dispersion control. The timing signals of micro-LED displays include frame VSYNC signals, horizontal synchronization (HSYNC) signals and the pixel clock signal CLK, which generates the PWM drive signal. The display period of a frame is divided into M subframes. The luminescence enable signal EM is re-broken based on the high- and low-bit widths of the display data. The high grey-scale refresh rate can be calculated as follows:

$$f_{high} = 60 \cdot M \cdot N \tag{4}$$

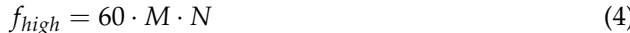
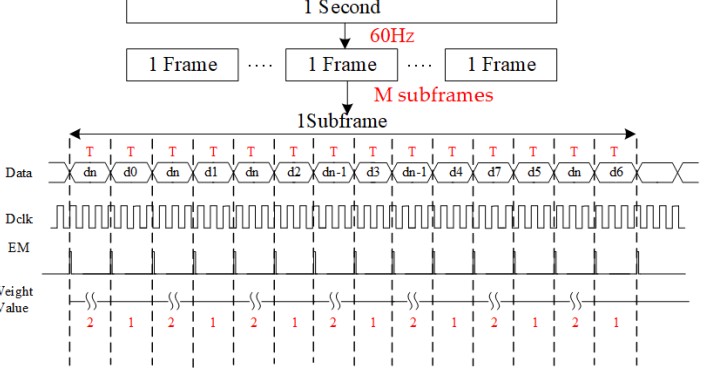

**Figure 10.** The grey-scale control model.

And the low grey-scale refresh rate can be expressed as follows:

$$f_{low} = 60 \cdot M \tag{5}$$

where M is the number of subframes, and the N is the number of groups after data are reallocated.

## 3. Experiment and Results

In this study, a glass-based micro-LED display was fabricated using a circuit made of LTPS TFT. The pitch is designed to be 0.6 mm, and the size of the LED is 76.2 × 152.4 μm. The current-to-voltage characteristics of RGB micro-LED are shown in Figure 11.

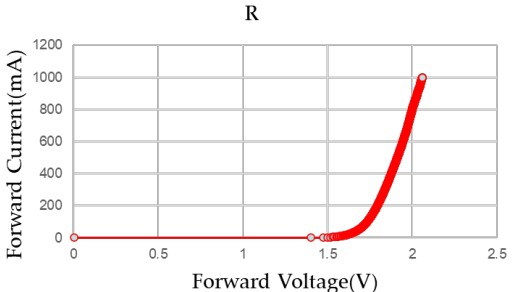

(**a**) R micro-LED

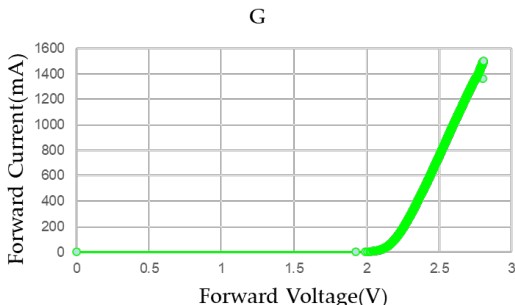

(**b**) B micro-LED

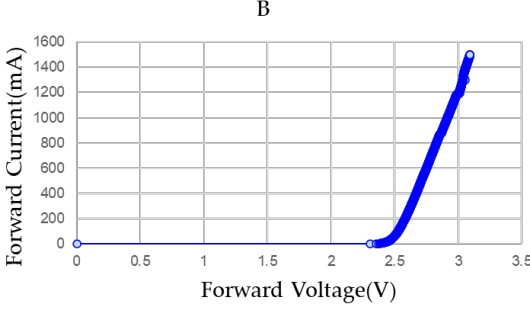

(**c**) G micro-LED

**Figure 11.** The current-to-voltage curves of RGB micro-LED.

The materials of red, blue and green LEDs are AlGaInP and InGaN/GaN. The voltage required for the R LED is approximately 1.8 V, which is lower than that of the G and B LED counterparts, which require 2.5 V. Then, the RGB micro-LED components are mounted on the panel with the mass-transfer machine, as shown in Figure 12.

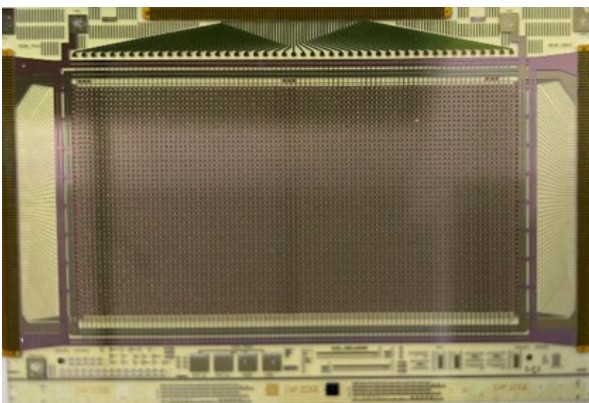

**Figure 12.** The micro-LED panel.

The control board is designed using four pieces of FGPA, DAC, ADC and SPDT, the power conversion chip and other components. In order to confirm the efficacy of the proposed method, we use the XC7A200T FPGA to present a new architecture to implement the micro-LED displays. The FPGA core board is shown in Figure 13. As the FPGA core board has access to a total of 180 I/O pins, and given that we require a total of approximately 730 I/O pins, we determine that we will require four FPGA core boards to both manage and operate the micro-LED displays effectively. We employ 10-bit DAC and ADC to obtain grey-scale voltage. With 1024 grey levels, the minimum driving voltage is about 7.8 mV. The supply voltage is 5 V, meaning that we use the power conversion chips to supply the FPGA power and panel power. Then, the micro-LED panel is connected to the control board through a flexible printed circuit, as shown in Figure 14.

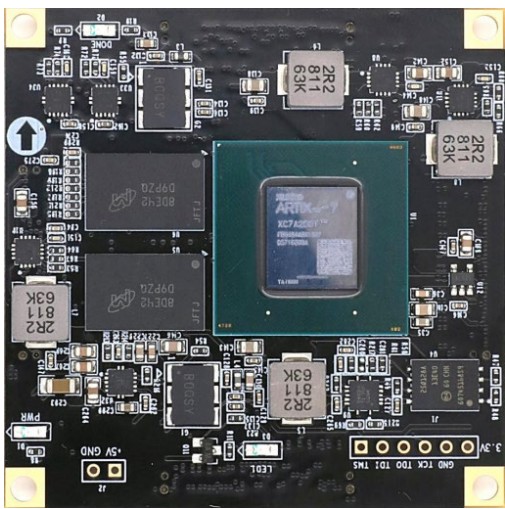

**Figure 13.** FPGA core board.

The control signals and 1–6 columns of RGB data for the micro-LED displays are generated by the master FPGA. The remaining columns of RGB data voltage (7–12, 13–18 and 19–24) are generated by the FPGA_2, FPGA_3 and FPGA_4, respectively, to enable complete control of the displays. Various control signals are obtained through FPGA programming according to the timing relationship in Figure 13. The simulation waveform of the whole control system was obtained through Verilog programming, as shown in Figure 15. The actual waveform that was obtained through an oscilloscope measurement is shown in Figure 16.

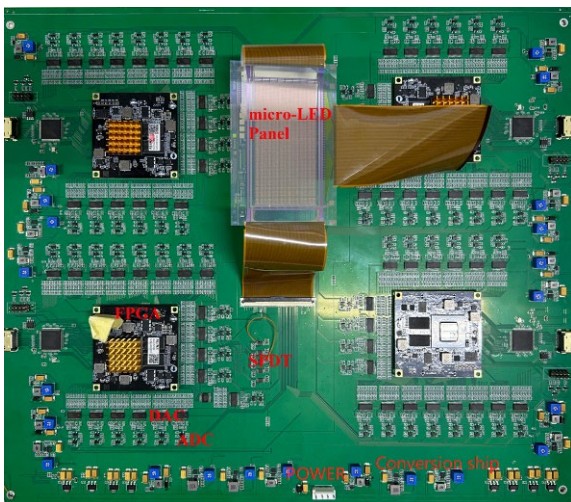

**Figure 14.** The control board.

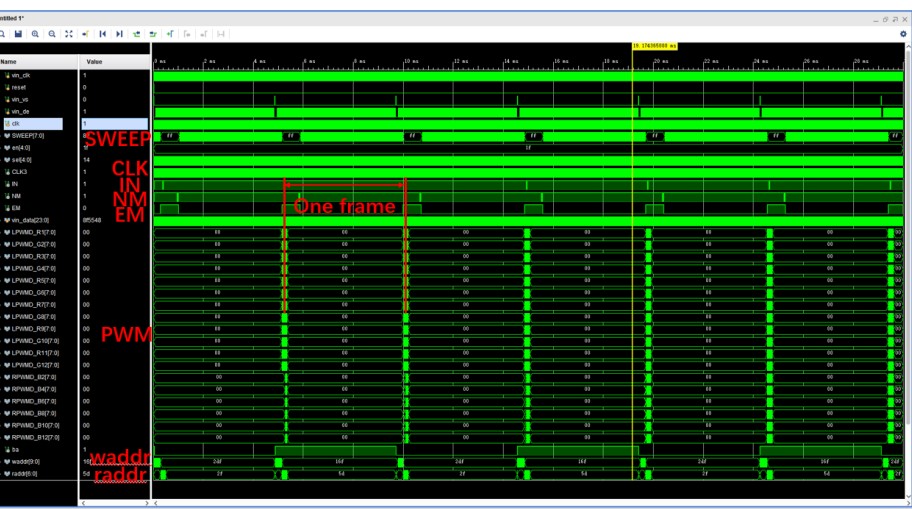

**Figure 15.** The simulation results.

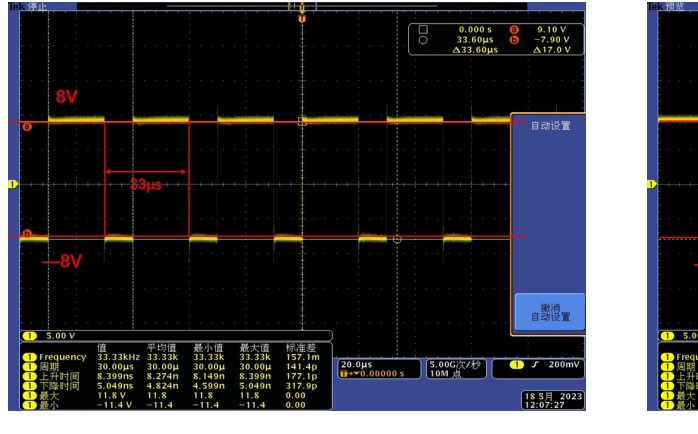

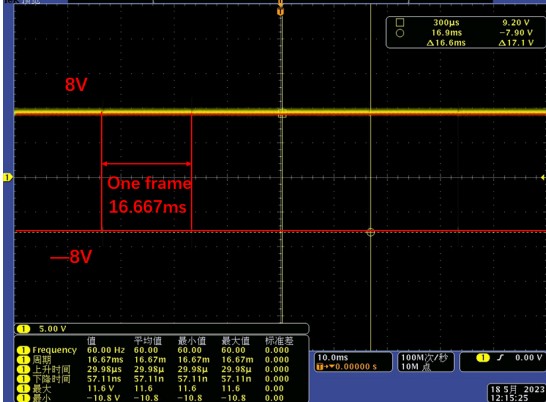

(**a**)                                                                                                          (**b**)

**Figure 16.** *Cont.*

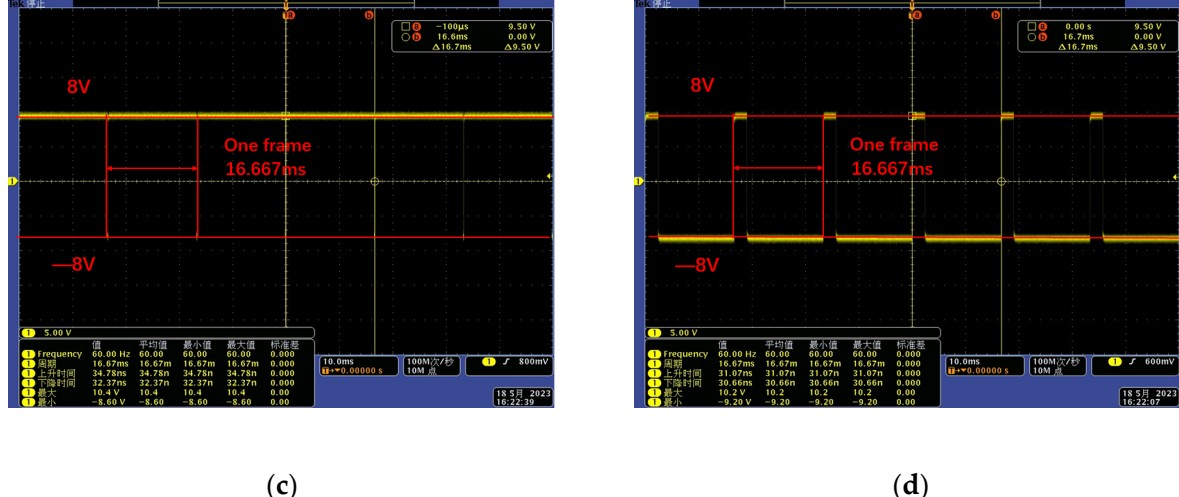

(**c**)                                    (**d**)

**Figure 16.** The frequency, period, rise time, fall time, maximum value, minimum value of (**a**) CLK, (**b**) IN, (**c**) NM and (**d**) EM.

The master FPGA generates signals with specific frequencies that are then transmitted to the SPDT. The SPDT is loaded with voltage to obtain various pulse signals, including CLK1, CLK2, IN, NM and EM. As can be seen from Equation (1), the display duration and grey levels of the display are dependent on the SWEEP's variation precision. Therefore, the 12-bit DAC is used in this work to increase the accuracy of the SWEEP's ramp performance. The SWEEP has a voltage range of 0 to 9 V divided into 4096 steps, with every step's amplitude reduced by 2.2 mV. While RGB data voltage vary from 0 to 8 V, which is divided into 1024 steps according to the 10-bit grey-scale, and the voltage variation in each step is 7.8 mV. Therefore, the SWEEP's precision meets the 10-bit RGB data output demand. The SWEEP and PWM obtained via oscilloscope measurement are shown in Figure 17.

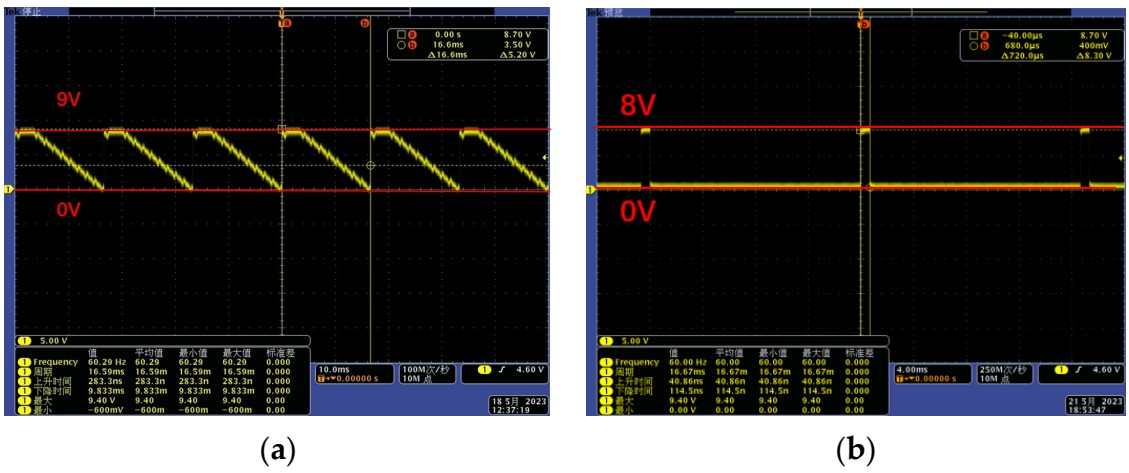

(**a**)                                    (**b**)

**Figure 17.** The frequency, period, rise time, fall time, maximum value, minimum value of (**a**) SWEEP and (**b**) PWM data.

To demonstrate the real-time system for micro-LED displays, the video is conveyed through HDMI to the control board. And then the video data are processed by FPGA and sent to the panel. Considering that the TFT writing time is about 3 μs and the regulation of time-slice dispersion from Figure 10, the values of N and M can be set as N = 7 and M = 5. From the Equations (4) and (5), it can be deduced that the high grey-scale refresh rate is 2100 Hz, and the low grey-scale refresh rate is 300 Hz. The power consumption of the 24 × 46 array is about 0.8 W. Figure 18 shows the picture of the micro-LED.

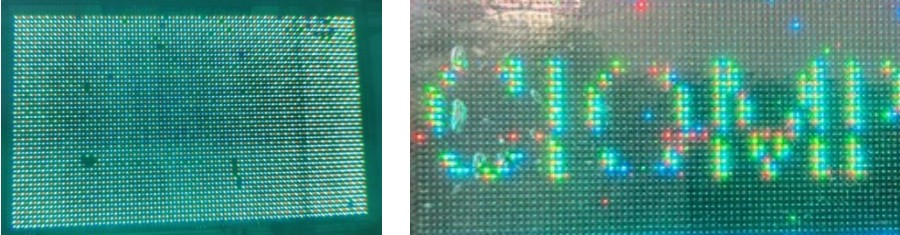

**Figure 18.** Display of micro-LED.

The screen brightness data are collected using a Charge-coupled Device (CCD) camera, and then the correction parameters CoefR, CoefG and CoefB are calculated, as shown in Figure 19. Considering that brightness correction results in grey-scale loss, we use front-end-by-front-end video correction that we used front-end correction.

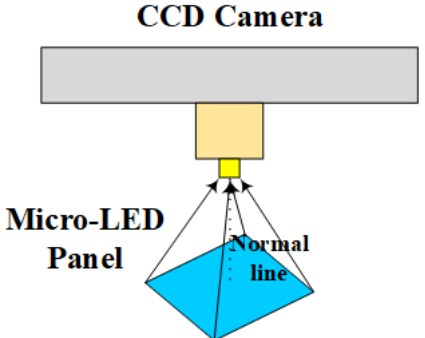

**Figure 19.** Brightness measuring equipment.

After conducting brightness correction, the nine-point method is used to measure the brightness values of the RGB at maximum brightness. The value is shown in Table 1. The brightness uniformity of RGB can be calculated using Equation (6).

$$uniformity = 1 - \frac{|L_i - L_{mean}|}{L_{mean}} \times 100\% \tag{6}$$

By substituting the values from Table 2 into Equation (6), the brightness uniformity of RGB is revealed to be 85.2%, 86.0% and 86.3%.

**Table 2.** The brightness measurement data after correction (cd/m$^2$).

| Measure | Red | Green | Blue |
|---|---|---|---|
| 1 | 539.9540 | 2024.5541 | 302.8855 |
| 2 | 480.2963 | 1863.8584 | 250.3257 |
| 3 | 443.4358 | 2002.9338 | 271.5751 |
| 4 | 555.7998 | 1940.3732 | 265.6749 |
| 5 | 481.4626 | 1817.2273 | 260.0550 |
| 6 | 453.2011 | 2037.6756 | 284.8278 |
| 7 | 507.1198 | 2206.0957 | 297.7589 |
| 8 | 460.3187 | 1746.6416 | 259.3232 |
| 9 | 411.1197 | 2033.2636 | 262.2923 |
| Uniformity | 85.2% | 86.0% | 86.3% |

The comparison with previous work is given in Table 3. It is shown that the refresh rate and brightness parameters are advanced in this paper.

**Table 3.** Comparison with previous works.

| Item | [12] | [13] | [14] | [16] | This Paper |
|---|---|---|---|---|---|
| High refresh rates | 60 Hz | 60 Hz | 384 Hz | 120 Hz | 2100 Hz |
| Resolution | 76 × BBB × 78 | 16 × GGG × 29 | 64 × RGB × 64 | 480 × RGB × 270 | 24 × RGB × 46 |
| Brightness (cd/m$^2$) | 367.2 | 970 | NA | 200 | 2706 |

## 4. Conclusions

We presented a new architecture based on the FPGA control system to realize the micro-LED displays. LTPS technology was employed to prepare the glass substrate, and the LEDs were soldered to the pads of the substrate. A 12-bit DAC was used to generate the high-precision SWEEP signal. Combining it with the 10-bit PWM, we achieved the 10-bit grey-scale micro-LED displays. We introduced a new grey-scale control model, which has enabled us to enhance the refresh rate to 2100 Hz in high grey-scale and 300 Hz in low grey-scale. The control system can provide a reference for micro-LED display control. Furthermore, we have proposed gamma correction and brightness correction methods to improve the uniformity of micro-LED displays up to 85%.

**Author Contributions:** Conceptualization, Y.C. (Yufeng Chen) and X.Z.; methodology, Y.C. (Yufeng Chen) and H.C. (Hui Cao); software (Vivado 2019.1), Y.C. (Yufeng Chen) and H.C. (Hui Cao); validation, Y.C. (Yufeng Chen) and H.C. (Hui Cao); formal analysis, Y.W.; investigation, H.C. (Hongbin Cheng); resources, Y.C. (Yu Chen); data curation, J.C. and S.H.; writing—original draft preparation, Y.C. (Yufeng Chen); writing—review and editing, Y.C. (Yufeng Chen); visualization, D.H. and J.L.; supervision, X.Z.; project administration, H.C. (Hui Cao). All authors have read and agreed to the published version of the manuscript.

**Funding:** This work was funded by the major science and technology special projects of the Jilin Province Science and Technology Development Program of China, grant number 20210301002GX.

**Data Availability Statement:** Not applicable.

**Conflicts of Interest:** Author Yufeng Chen, Xifeng Zheng, Hui Cao, Yang Wang, Hongbin Cheng, Yu Chen were employed by the company Changchun Cedar Electronics Technology Co., Ltd. The remaining authors declare that the research was conducted in the absence of any commercial or financial relationships that could be construed as a potential conflict of interest.

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
