# Peer review of "High Precision Control System for Micro-LED Displays"

_applsci, doi:10.3390/app131910601_

Round 1

Reviewer 1 Report

The authors present a new architecture for an extended Micro LED display based on FPGA control system, using low temperature polysilicon technology on the glass substrate. There is good writing and fluency. I think that the detailed research and results are publishable for this journal and that this study can be referenced in the future.

1-The doi extension of reference No. 2 is missing. Authors are advised to double-check references. 

2-It would be beneficial for the readers if the authors provided a table containing a comparison of 'the micro LEDs they produced' and 'the high refresh rates they achieved' with examples from the literature.

It is suggested that this study can be published after the recommendations of other reviewers are completed.

Author Response

Comments 1: The doi extension of reference No. 2 is missing. Authors are advised to double-check references.
Response 1: I have reviewed the references and updated the doi for reference No. 2 and No. 10.Thank you for pointing this out. I agree with this comment. Therefore, I have reviewed the references and updated the doi for reference No. 2 and No. 10. You can see these change in References, page12 and page 13.

Comments 2: It would be beneficial for the readers if the authors provided a table containing a comparison of 'the micro LEDs they produced' and 'the high refresh rates they achieved' with examples from the literature.
Response 2: I have prepared a table that outlines the high refresh rates, resolution, and brightness comparisons between the prior studies and this current research. You can see these change in page12, table3.

Reviewer 2 Report

This paper presents a new control system for micro-LED display based on pulse width modulation of voltage. The authors claimed that the proposed method can achieve a higher refresh rate and a better display uniformity, compared with the conventional current control scheme. I feel the paper is publishable in Applied Sciences, but some revisions are required.  

1)     The authors discussed in the introduction that one of the problems of the current amplitude control method is the existence of color casting issue resulting from threshold voltage drift of TFT and brightness fluctuation of micro-LEDs. However, this issue is also present in the proposed control system as discussed in page 5, and data correction is required to achieve a uniform display brightness. I’m a little bit confused with the advantages of the proposed method as compared with the direct current control method. The advantages of the proposed method as compared with the direct current modulation method should be discussed more explicitly.

2)     Please give the materials for the red, blue, and green LEDs (page 7, 8). Red LED should be AlGaInP. Blue and green LEDs should be InGaN/GaN. Is this correct?

3)     Please add scales (voltage, time) to Figs. 15, 16, 17. Letters in the screen shot of the oscilloscope are too small to read.

4)     The brightness of the fabricated micro-LED display seems to be very low. Is the brightness limited by the driving circuit? Please give the driving conditions (injection current or power consumption) used for brightness measurements.   

 The manuscript contains numerous grammar mistakes and typos. Please check the language carefully.  

Author Response

Comments 1: The authors discussed in the introduction that one of the problems of the current amplitude control method is the existence of color casting issue resulting from threshold voltage drift of TFT and brightness fluctuation of micro-LEDs. However, this issue is also present in the proposed control system as discussed in page 5, and data correction is required to achieve a uniform display brightness. I’m a little bit confused with the advantages of the proposed method as compared with the direct current control method. The advantages of the proposed method as compared with the direct current modulation method should be discussed more explicitly.
Response 1: The direct current modulation method may case color casting issue resulting from threshold voltage drift of TFT. The brightness fluctuation of micro-LED is cause by brightness differences between micro-LED and IR drop. We propose using gamma correction and brightness correction to improve the uniformity of micro-LED display to 85%. The proposed method in this paper can provide a digital-analog hybrid method, which can combine traditional direct current modulation method and PWM driving to achieve constant current drive control. You can see these change in page 2, paragraph 1 and page 5, paragraph 5.

Comments 2: Please give the materials for the red, blue, and green LEDs (page 7, 8). Red LED should be AlGaInP. Blue and green LEDs should be InGaN/GaN. Is this correct?
Response 2: I have replenished the materials for the red, blue, and green LEDs. Professor, you are right. The materials of Red, Blue and green LEDs are AlGaInP and InGaN/GaN. You can see these change in page7, paragraph 5.

Comments 3: Please add scales (voltage, time) to Figs. 15, 16, 17. Letters in the screen shot of the oscilloscope are too small to read.
Response 3: I have added scales (voltage, time) to Figs. 15, 16, 17, as shown in the paper. You can see these change in page9, and page 10.

Comments 4: The brightness of the fabricated micro-LED display seems to be very low. Is the brightness limited by the driving circuit? Please give the driving conditions (injection current or power consumption) used for brightness measurements.
Response 4: The white brightness of the panel in this paper is 2706 cd/m2 as shown in talbe2. The table 3 shows that the brightness parameters are advanced in this paper. The circuit's driving capacity limits the brightness. The TFT current formula demonstrates that the flow of current through the LEDs is restricted by the W and L, which represent the channel width and length. The material and process restrictions make it impossible to increase the width-to-length ratio indefinitely. The power consumption of the 24×46 array is about 0.8W. You can see these change in page11 paragraph 1.

Response to Comments on the Quality of English Language
Point 1: The manuscript contains numerous grammar mistakes and typos. Please check the language carefully.
Response 1: Some changes were made to address grammatical issues in the paper, such as the highlighted section of the paper.

Round 2

Reviewer 2 Report

The authors have properly addressed my comments. I think that the manuscript is almost acceptable for publication, but a few minor revisions are required. 

1)     Figure 11, “current-to-voltage curves” should be “current versus voltage curves” or “current-voltage curves”.

2)   Line 164, “The resolution of the micro-LED is 24×46” should be “The pixel number of the micro-LED is 24×46”. “Resolution” is the number of pixels contained in one inch.  

The following language mistakes should be corrected.

1)     Line 212, “presented” should be “present”.

2)  Line 73, “are is” should be “are”.

Author Response

Point-by-point response to Comments and Suggestions for Authors.

Comments 1: Figure 11, “current-to-voltage curves” should be “current versus voltage curves” or “current-voltage curves”.
Response 1: The caption for Figure 11 has been amended and is visible in line 207.

Comments 2: Line 164, “The resolution of the micro-LED is 24×46” should be “The pixel number of the micro-LED is 24×46”. “Resolution” is the number of pixels contained in one inch.
Response 2: I have made a change in line 164.

Response to Comments on the Quality of English Language.

Point 1: The following language mistakes should be corrected.
1) Line 212, “presented” should be “present”.
2) Line 73, “are is” should be “are”.
Response 1: The above two errors have been fixed and can be seen in lines 73 and 212.
